# Membrane Insertion of the M13 Minor Coat Protein G3p Is Dependent on YidC and the SecAYEG Translocase

**DOI:** 10.3390/v13071414

**Published:** 2021-07-20

**Authors:** Farina Kleinbeck, Andreas Kuhn

**Affiliations:** Institute of Biology, University of Hohenheim, 70599 Stuttgart, Germany; Farina_Kleinbeck@uni-hohenheim.de

**Keywords:** bacteriophage M13, membrane protein insertion, translocase SecYEG, membrane insertase YidC, membrane potential, phage assembly, disulphide crosslinking, ribosome-nascent chain

## Abstract

The minor coat protein G3p of bacteriophage M13 is the key component for the host interaction of this virus and binds to *Escherichia coli* at the tip of the F pili. As we show here, during the biosynthesis of G3p as a preprotein, the signal sequence interacts primarily with SecY, whereas the hydrophobic anchor sequence at the C-terminus interacts with YidC. Using arrested nascent chains and thiol crosslinking, we show here that the ribosome-exposed signal sequence is first contacted by SecY but not by YidC, suggesting that only SecYEG is involved at this early stage. The protein has a large periplasmic domain, a hydrophobic anchor sequence of 21 residues and a short C-terminal tail that remains in the cytoplasm. During the later synthesis of the entire G3p, the residues 387, 389 and 392 in anchor domain contact YidC in its hydrophobic slide to hold translocation of the C-terminal tail. Finally, the protein is processed by leader peptidase and assembled into new progeny phage particles that are extruded out of the cell.

## 1. Introduction

The M13 G3p is synthesised with an 18-amino-acid-residue-long signal sequence, a large periplasmic domain of 379 residues, a 21-residue-long membrane anchor domain and a C-terminal tail of six residues located in the cytoplasm (Figure 1). This allows the newly synthesised protein to insert into the inner membrane of *Escherichia coli* before its assembly into the phage progeny particles [1]. After processing, the periplasmic part of G3p folds into β-structured N1 and N2 domains that are separated by glycine-rich regions (L1 and L2) from CT, encompassing the C1 and C2 domains [2,3]. The membrane anchoring of G3p has been studied in detail [4,5] and these pioneering studies defined a membrane anchor region. The membrane-spanning protein is assembled onto the proximal end of the phage particles that are extruded from the infected cells. On the phage, the C-terminal tail is anchored to the phage capsid, whereas N1 and N2 are exposed at the surface of the phage particle. G3p has been extensively used for phage display technology [6]. The random sequences encoding the display peptides are introduced between the signal peptide and the mature sequence of G3p. Therefore, the peptides are displayed on the phage at the very N-terminus, not affecting the two β-structured domains, N1 and N2.

For infecting a host cell, the N2 domain first binds to the tip of the F-pilus. The binding of the N2 domain to the F-pilus causes a conformational change in G3p which exposes the TolA-binding site of the N1 domain. The pilus retracts and drags the phage into the periplasm where the N1 domain contacts TolA [7,8]. The phage particle is then dissociated, the coat proteins (G3p, G6p, G8p, G7p and G9p) partition into the inner membrane of *E. coli* and the single-stranded DNA is released from the coat proteins and delivered into the cytoplasm for replication and gene expression. How these events occur at the molecular level is presently unknown.

Here, we were interested in how the newly synthesised G3p is targeted and inserted into the inner membrane. Most bacterial membrane proteins are inserted by the SecYEG translocase together with the associated components SecA and YidC [9]. Recent research has uncovered the molecular structures of all the involved components with atomic resolution [10,11,12,13]. How these translocation components function and actually interact with the substrates before partitioning into the membrane is largely unknown. Some recent studies have defined arrested intermediate stages of inserting proteins that show the nascent protein chain within the SecY translocation tunnel [12,14,15,16] for Sec-dependent proteins, and the hydrophobic slide of YidC [17,18,19] for YidC-dependent proteins. However, as shown here, the M13 coat protein G3p requires both SecYEG and YidC for its membrane insertion, and we were interested to know when it contacts either component at the different stages of its synthesis.

**Figure 1 viruses-13-01414-f001:**
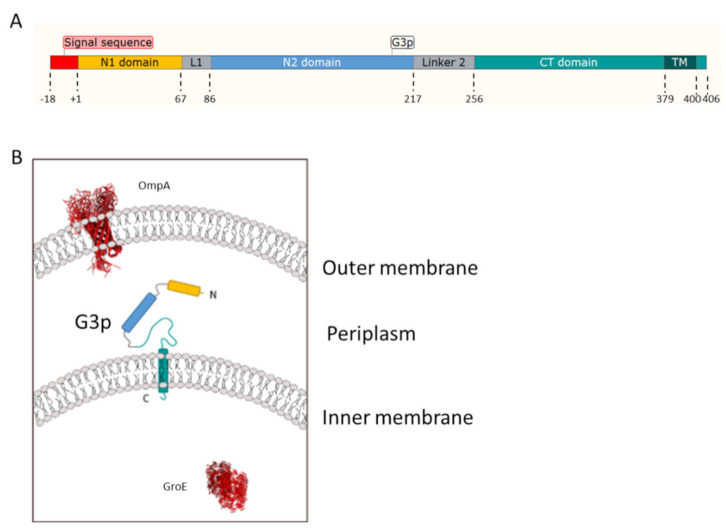
Structural features and topology of G3p. (**A**) Schematic representation of the G3p protein regions with the signal sequence (red, residues −18 to −1), the N1 domain (yellow, residues 1 to 67), the linker 1 region (grey), the N2 domain (blue, residues 86 to 217), the linker 2 region (grey) and the CT domain (green, residues 256 to 406), which includes the membrane anchor segment (380 to 400) and the cytoplasmic domain (residues 401 to 406). (**B**) Topology of the G3p protein domains in the periplasm and spanning the inner membrane using the same colour code as in (**A**). Depicted are the outer membrane OmpA and the cytoplasm GroE (in red) that were used in the localisation studies as control proteins.

To study the membrane targeting and membrane insertion of G3p, we followed here the newly synthesised protein in *E. coli* strains that lack the mostly essential components. For membrane targeting, these are the well-known components SecB or the signal recognition particle SRP. To study the membrane insertion process, the involvement of SecA, SecYE and YidC was tested accordingly. Direct interaction with these components was further studied by disulphide crosslinking experiments, showing that the signal sequence of G3p primarily interacts with SecYE and the C-terminal membrane anchor region with YidC.

## 2. Materials and Methods

### 2.1. Bacterial Strains and Plasmids

YidC depletion was accomplished using the MK6 stain where the chromosomal YidC promoter was replaced by the *araBAD* promoter. This allows the depletion of the chromosomal YidC by growth in 0.4% glucose medium. MK6 is a MC1061 derivative: *F^-^ araD 139 Δ (argF-lac) U169 rspL150 rel A 1 flb B5301 fruA 25 deo C1 ptsF25 ParaBA-YidC* [20].

SecE depletion was accomplished using the MG1655-SecE (pCP20) which is a derivative of GS0479 [21] where the chloramphenicol resistance cassette was removed by pCP20. The chromosomal SecE gene is under the control of P*araBAD*. The depletion of SecE results in a rapid loss of SecY [22].

Ffh depletion strain MC-dFfh has the *ffh* gene under the control of the araBAD promoter. MC-dFfh is a MC1061 derivative [23].

SecB deletion strain JW3584 was from the Keio collection [24]. *E. coli* strain MC1061 was used for the SecA inactivation via sodium azide and as a positive control for the SecB deletion strain JW3584 [25].

M13 G3p was expressed from pMS-g3p containing *gene3* amplified from the M13 genome by the forward primer containing a flanking 5′ *Ase*I restriction site (5′ GGCATTAATGAAAAAATTATTATTCGCAATTCC3′) and reverse primer containing a flanking 3′ *Hind*III restriction site (5′ GCAAGCTTAGACTCCTTATTACGC 3′). The PCR product was first cloned into pET22b and then subcloned into pMS119EH [26] using restriction sites 5′ *Xba*I (from the MCS of pET22b) and 3′ *Hind*III. All *E. coli* strains for the proteinase K mapping were transformed with pMS-g3p and grow in the presence of 100 µg/mL ampicillin. For the disulphide crosslinking studies, all 8 endogenous cysteines of G3p were replaced with serines for pMS-g3pC0 and in the signal sequence the serine at position -3 was replaced with a phenylalanine, resulting in pMS-g3pH5C0 using the following primers: 5′CTGTTGAAAGTAGCTTAGCAAAACC, 5′CTAACTATGAGGGTAGCCTGTGGAATGCTAC, 5′CAGGCGTTGTAGTTAGCACTGGTGACGAAAC, 5′ GACGAAACTCAGAGCTACGGTACATG, 5′GTAAATTCAGAGACAGCGCTTTCCA-TTC, 5′GATCCATTCGTTAGCGAATATCAAGG, 5′CAATCGGTTGAAAGCCGCCC-TTTTGTC, 5′CATATGAATTTTCTATTGATAGCGACAAAATAAACTTATTCC. Site-directed mutagenesis was performed on pMS-g3pH5C0 to introduce single cysteines at the following positions, respectively: F-13C: 5′GAAAAAATTATTATGCGCAATTCCTTTAG; A-12C: 5′GAAAAAATTATTATTCTGTATTCCTTTAGTTGTTC, L-9C: 5′CGCAATT--CCTTGTGTTGTTCCTTTCTATTC, V-8C: 5′CGCAATTCCTTTATGTGTTCCTTTCTATTC and on pMS-g3pC0: L385C: 5′GGTGTCTTTGCGTTTCTTTGTTATGTTGCCACCTTTATG, L386C: 5′GTCTTTGCGTTTCTTTTATGTGTTGCCACCTTTATGTATG, L387C: 5′CGTTTCTTTTATATTGTGCCACCTTTATG, L388C: 5′GCGTTTCTTTTATATGTTTGCACCTTTATGTATG, L389C: 5′CTTTTATATGTTGCCTGCTTTATGT-ATGTATTTTC, L390C: 5′CTTTTATATGTTGCCACCTGTATGTATGTATTTTC, L391C: 5′GTTGCCACCTTTTGTTATGTATTTTCTACG, L392C: 5′GCCACCTTTATGTGTGTATTTTCTACG, L393C: 5′GCCACCTTTATGTATTGTTTTTCTACGTTTGC.

For disulphide crosslinking with G3p RNCs, the G3p constructs G3nc8 and G3nc35 were amplified from pMS-g3pH5F-13C, A-12C, L-9C and V-8C by a forward primer containing a flanking 5′ *Eco*RI restriction site (5′GATAGGAATTCATAATTTTGTTTA ACTTTAAGAAGGAG) and the respective reverse primer with a flanking 3′ *Xba*I restriction site (G3nc8: 5′GATAGGAATTCATAATTTTGTTTAACTTTAAGAAGGAG; G3nc35: 5′TGCTCTAGAACCCTCATAGTTAGCGTAACG). The Flag-tag and SecM stalling sequence was amplified from pMS119Pf3-P2 [17] by a forward primer containing a flanking 5′ *Xba*I restriction site (5′TGCTCTAGAGACTACAAGGACCAC GAC) and reverse primer containing a flanking 3′ *Hind*III restriction site (5′CCCAAG CTTAGGGCTACCACGGATTG). The PCR products were then cloned into pMS119EH using restriction sites 5′ *Eco*RI and 3′ *Xba*I and 5′ *Xba*I and 3′ *Hind*III, respectively.

For disulphide crosslinking with SecY, the sequence for *secYEG* was amplified from pTrc99a-T7-tagSecY (C0) EG [22] by a forward primer containing a flanking 5′ *Xba*I site and a ribosome-binding site (5′GCTCTAGAAGGAGATAAGATCTATGGCGTCCATG-ACCGGCGG) and reverse primer containing a flanking 3′ *Hind*III restriction site (5′GGCAAGCTTTTAGTTCGGGATATCGCTGGTCGGC). Site-directed mutagenesis was performed to introduce singe cysteine mutants at the following positions: S68C: (5′GTTTAACATGTTCTGTGGTGGTGCTCTC) and S69C: (5′CATGTTCTCTTGTGGTG-CTCTCAGC). For the disulphide crosslinking studies with Ffh, pMS-Ffh-C406S-C-Strep and pMS-Ffh-M423C-C-Strep [27] were subcloned into pGZ119EH [26] using restriction sites 5′ *Nco*I and 3′ *Hind*III. The YidC constructs pGZ-YidC-L427C, L434C, I501C, F505C, W508C are described [20].

### 2.2. Proteinase K Mapping

For the depletion strains MK6, MC-dFfh or MG1655-SecE(pCP20), respectively, overnight cultures were inoculated in LB media containing 100 µL/mL ampicillin, 0.2% arabinose and 0.4% glucose at 37 °C. Then, the cells were washed twice with LB media and back diluted 1:50 in fresh medium, with either 0.4% glucose and 0.2% arabinose to maintain the chromosomal YidC at a reasonable level or grown in only 0.4% glucose to inhibit the expression of the gene, resulting in the depletion of the respective protein. To ensure full depletion, the cells were grown for 3 h at 37 °C until an OD_600_ of 0.5 was reached. For each condition, a sample was precipitated with TCA and analysed for a Western blot to ensure expression and depletion of the respective protein. For the protease mapping, 1 mL of each culture was washed twice with M9 salt solution (39 mM Na_2_HPO_4_, 22 mM KH_2_PO_4_, 8 mM NH_4_Cl, 18 mM NaCl) and subsequently resuspended in M9 media (39 mM Na_2_HPO_4_, 22 mM KH_2_PO_4_, 8 mM NH_4_Cl, 18 mM NaCl, 1 mM MgSO_4_, 0.1 mM CaCl_2_, 5 µg/mL thiamine, 0.0005% ammonium ferric citrate) containing 18 amino acids, but methionine and cysteine supplemented with either 0.4% glucose or 0.2% arabinose, respectively. The cells were grown for 1 h at 37 °C, induced with 1 mM IPTG for 10 min, pulse labelled with 30 µCi ^35^S-l-methionine/cysteine for 3 min and chased with 250 µg/mL L-methionine/cysteine for 1 min. Then, cells were spun down and resuspended in 600 µL spheroplast buffer (40% sucrose, 33 mM Tris-HCl pH 8), treated with 5 µg lysozyme and 0.5 mM EDTA and incubated on ice for 10 min. The generated spheroplasts were kept on ice and split into 3 aliquots of 200 µL each, where one was treated immediately with 10% TCA, the second with 0.2 mg proteinase K and the third with 0.2 mg proteinase K and 1% Triton-X-100 for 1 h on ice, followed by precipitation with 10% TCA.

### 2.3. Immunoprecipitation

The TCA-precipitated samples were washed with ice cold acetone, dried, resuspended with 50 µL 2% SDS and 100 mM Tris-HCl (pH 8) buffer and boiled (95 °C) for 5 min. Then, 1 mL TEN-TX buffer (150 mM NaCl, 10 mM Tris-HCl pH 8, 1 mM EDTA, 1% Triton-X-100) and 20 µL StaphA solution (USBiological Life Science, Swampscott, Mass., USA) was added. The preclearing of the samples was carried out at 4 °C on a rotary wheel for 1 h, the StaphA was spun down and the supernatant was split into three aliquots. Respectively, antibody was added to detect G3p, GroEL and OmpA and kept overnight at 4 °C. Then, the samples were spun down, washed twice with 1 mL TEN-TX and once with TEN buffer (150 mM NaCl, 10 mM TrisHCl pH 8, 1 mM EDTA). SDS-PAGE sample buffer was added, boiled for 5 min, spun down and applied on 12% SDS-PAGE for samples treated with antibody to detect G3p and 14% SDS-PAGE for GroEL and OmpA. The labelled protein bands were detected by phosphorimaging.

### 2.4. Disulphide Crosslinking

For disulphide crosslinking of G3p and YidC, MK6 cells were cotransformed with the respective pGZ-YidC encoding a single cysteine mutant and with pMS-g3p encoding a single cysteine mutant in the signal peptide or in the transmembrane segment, respectively, as described in Section 2.1. For disulphide crosslinking between SecY and G3p, MG1655-SecE cells were cotransformed with pGZ-SecYI408C and pMS-g3p encoding a single cysteine mutant in the transmembrane segment and pGZ-SecYS68C or pGZ-SecYS69C with both G3p RNC constructs, respectively. For disulphide crosslinking with Ffh and the G3p RNC constructs, MC-dFfh cells were cotransformed with pGZ-Ffh-C406S-C-Strep or pGZ-Ffh-M423C-C-Strep and both G3p RNC constructs, respectively.

For the depletion strains MK6 and MC-dFfh, MG1655-SecE(pCP20) overnight cultures were inoculated in LB media containing 100 µL/mL ampicillin, 0.2% arabinose and 0.4% glucose at 37 °C. Then, cells were washed twice with LB media and then back diluted 1:50 in fresh medium and supplemented with 0.4% glucose to inhibit the expression of the respective gene that causes depletion of the protein YidC in MK6, SecE in MG1655-SecE(pCP20) and Ffh in MC-dFfh. To ensure full depletion, the cells were grown for 3 h at 37 °C until an OD_600_ of 0.5 was reached. Then, 400 µL of each culture was washed twice with M9 salt solution (39 mM Na_2_HPO_4_, 22 mM KH_2_PO_4_, 8 mM NH_4_Cl, 18 mM NaCl) and subsequently resuspended in M9 medium (39 mM Na_2_HPO_4_, 22 mM KH_2_PO_4_, 8 mM NH_4_Cl, 18 mM NaCl, 1 mM MgSO_4_, 0.1 mM CaCl_2_, 5 µg/mL thiamine, 0.0005% ammonium ferric citrate) containing 18 amino acids, but methionine and cysteine supplemented with 0.4% glucose. The cells were grown for 1 h at 37 °C, induced with 1 mM IPTG for 10 min, pulse labelled with 15 µCi ^35^S-l-methionine/cysteine for 2 min and then supplemented with 1 mM freshly mixed copper phenanthroline for 10 min. The samples were TCA precipitated, and immunoprecipitated (as described in Section 2.3) with antibody to YidC, the T7-tag for SecY detection and Flag-tag for G3p RNC detection. If not indicated otherwise, samples were prepared for SDS-PAGE in a DTT-free sample buffer. A 10% SDS-PAGE was used for disulphide crosslinking with YidC, 12% for SecYI408C and 14% with SecY and G3p RNC constructs and Ffh and G3p RNC constructs.

## 3. Results

### 3.1. Membrane Targeting of G3p

Most membrane proteins are targeted by the signal recognition particle SRP, consisting of the Ffh protein and a 4.5S RNA in *Escherichia coli*. Most secreted proteins use SecB that interacts with SecA for translocation. To study the requirements of G3p for membrane targeting and insertion, we cloned *gene 3* into an expression plasmid and transformed *E. coli* cells that were conditionally defective for the known targeting and insertion components. First, the *E. coli* strain JW3584 that has the *secB* gene deleted was tested for G3p membrane insertion. The JW3584 (SecB^-^) or MC1061 (SecB^+^) cells bearing the plasmid pMS-g3p encoding G3p were grown to the exponential phase, induced for 10 min with IPTG and pulse labelled with ^35^S-methionine/cysteine for 3 min and chased with non-radioactive methionine/cysteine for 2 min. The cells were then spun down and converted to spheroplasts. Proteinase K was added to the outside of the spheroplasts to digest the translocated proteins. Immunoprecipitation with antiserum to M13 was followed by PAGE and fluorography. Regardless of whether SecB was present in the MC1061 cells or absent in the JW3584 cells, G3p was translocated and readily digested by the proteinase K (Figure 2A, lane 2 and 5). Since SecB is also required for the translocation of proOmpA, the cells were also analysed by immunoprecipitation with anti-OmpA serum (Appendix A). When SecB was deleted, the translocation and cleavage of proOmpA was retarded.

The SRP dependence was tested with the conditional MC-dFfh strain that has the Ffh expression under the control of the araBAD promoter. When the cells were grown in the presence of arabinose, Ffh was expressed, whereas in the absence of arabinose, Ffh was depleted (Figure 2D). Membrane insertion of G3p was analysed by Proteinase K mapping as described above. In the presence or absence of Ffh, G3p was capable of inserting into the membrane, exposing the antigenic N1 and N2 domains in the periplasm (Figure 2C, lanes 2 and 5). We conclude from these results that neither targeting factor SecB and SRP are required for G3p. However, it remains possible that G3p can use either SecB or SRP for membrane targeting.

### 3.2. Binding of G3p Nascent Chains to Ffh

Since our deletion and depletion strains did not provide us with a clear answer, we investigated whether G3p contacts Ffh early during its synthesis by disulphide crosslinking. To test this, we used a Ffh mutant where the methionine at position 423 in the signal sequence pocket was substituted with a cysteine [28]. In G3p, single cysteine mutations were introduced at several positions in the signal sequence followed by an added Flag-tag. The translation was arrested by an introduced *secM* stalling sequence to allow the exposure of the nascent chain from the ribosome (Figure 3A). When G3nc35 with a cysteine at −12 was coexpressed with Ffh423C and immunoprecipitated with the Flag-tag antibody, an additional band appeared that was recognised by the Flag-tag antibody (Figure 3B). This band disappeared when the sample was treated with dithiothreitol (+DTT, Figure 3C), indicating a disulphide bond between G3p and Ffh. Taken together, our results suggest that G3p interacts with Ffh for its targeting to the membrane. However, when Ffh is depleted targeting and membrane insertion of G3p still occurs, possibly with the help of SecB and SecA.

### 3.3. Membrane Insertion of G3p Requires SecAYEG Translocase and YidC

The involvement of the Sec translocase was first tested with sodium azide, which inhibits the SecA component [29] (Appendix A). Exponentially growing MC1061 cells expressing G3p were treated with 1 mM IPTG for 5 min and with 10 mM sodium azide for another 5 min, pulse labelled with ^35^S-methionine/cysteine for 3 min and chased for 2 min. The cells were converted to spheroplasts and proteinase K was added to the outside for 1 h. Without azide, the G3p protein was readily digested by the protease (Figure 4A, lane 2), whereas in the azide-treated cells G3p was inhibited for translocation and resistant to the protease (lane 5). The requirement of SecYEG was tested in the SecE depletion strain MG1655, where the expression of SecE and hence SecY is under the control of the *araBAD* promoter (Fontaine et al. 2011). When the cells were grown in the presence of arabinose and expressed SecYEG, G3p was translocated and accessible to the proteinase K (Figure 4B, lane 2). When the cells were grown without arabinose and the expression of SecYEG was inhibited, G3p translocation was not observed (lane 5).

The requirement of the membrane insertase YidC was tested in MK6, which also has an *araBAD* promoter in front of the chromosomal *yidC* gene [20]. In the absence of YidC G3p was not accessible to the proteinase K and therefore accumulated in a non-translocated state (Figure 4C, lane 5), showing that YidC is required for the membrane insertion of G3p. Depletion of YidC in the cells was verified by a Western blot (Appendix A).

We then tested whether the electrochemical membrane potential is required for the translocation of G3p and treated the MC1061 cells with the protonophor CCCP 45 s prior to pulse labelling. This led to an accumulation of the non-translocated G3p (Figure 4D, lane 5). For a control, we also tested the insertion and cleavage of proOmpA, which showed the accumulation and inhibition of its cleavage to OmpA (Appendix A). Taken together, our results show that the membrane insertion of G3p requires SecA, SecYEG, YidC and the membrane potential across the inner membrane.

### 3.4. Binding of G3p Nascent Chains to SecY and YidC

To investigate whether the SecYEG translocase and the YidC insertase are contacted by G3p early after its synthesis, we employed ribosome-nascent chains (RNCs) where only a short amino-terminal portion of G3p is exposed at the exit tunnel of the ribosome. To this end, a *secM* stalling sequence was introduced into *gene3* at different positions. In G3nc8, a nascent chain of 36 amino acid residues should be exposed outside the ribosome, whereas in G3nc35, a peptide of 63 residues is exposed, assuming that 40 residues are still inside the ribosome (Figure 3A). When the G3nc8 was coexpressed with SecY that contained a cysteine residue in the plug domain at position 68 or 69, a crosslink product was observed to the signal sequence of G3p at the residue −12 (Figure 5A,B). Similarly, with G3nc35, disulphide contact was observed at the same positions (Figure 5C,D). The testing of a possible contact to YidC was done by coexpressing the YidC single cysteine mutants at 427, 434, 501 and 508. No clear crosslinked band was observed, as shown exemplarily with residue 434 and 508 in the hydrophobic slide of YidC (Appendix A).

### 3.5. Binding of the G3p Membrane Anchor to SecY and YidC

Since both SecYEG and YidC are involved in the membrane insertion process of G3p early on, it was of interest to know whether both proteins also contact the membrane anchor region of G3p. Single cysteines were placed at the positions 385 to 393 in the centre of the membrane anchor region (380–400) and coexpressed with SecY mutants that have a unique cysteine residue at various positions in the translocation tunnel. The immunoprecipitations of the pulse-labelled cells showed no contacts to the residues 68, 97 and 275 of TM 2a, 2b and TM7, respectively, but did show a contact to the 408C residue of TM10 of SecY (Figure 6A). Strong contacts were observed with YidC at positions 427 and 505 (Figure 6B–D, Appendix A). These results suggest that the membrane anchor of G3p is mainly inserted by YidC. In conclusion, G3p membrane insertion is mainly initiated by the interaction of the signal peptide to SecY (Figure 7A) and terminated by the interaction of the membrane anchor region to YidC (Figure 7B).

## 4. Discussion

The minor coat protein G3p of bacteriophage M13 is a transient membrane protein that has an 18-residue-long signal sequence and a transmembrane anchor sequence close to the C-terminus of the protein. As for most membrane proteins of *E. coli*, it is expected that the newly synthesised protein first interacts with the SRP particle that guides the RNC complex to the membrane and the Sec translocase complex [30,31]. However, having a signal sequence and a large periplasmic region, G3p is more like a secretory bacterial protein which usually requires SecB as a chaperone, that subsequently interacts with SecA at the inner membrane surface [32,33,34]. We were therefore interested in how G3p finds its way to the membrane and first studied the targeting of the protein in strains that were deficient for SRP or SecB. To our surprise, neither deficiency prevented G3p from membrane insertion which can be explained by either the involvement of an additional targeting mechanism or both SRP and SecB targeting systems functioning for G3p mutually. Indeed, our attempt to accumulate nascent chains of G3p in the cells by introducing a *secM* stalling sequence [35] revealed distinct crosslinked products of the nascent chains with the Ffh protein, the major component of the bacterial SRP [36]. This result suggests that SRP is targeting G3p to the membrane. In case of its absence, however, another system might substitute for the loss of SRP function, possibly SecB or SecA. To test SecB interaction, we purified and partially unfolded G3p with 3.8 M urea and then incubated the protein for 1 h with purified SecB. At this urea concentration, the folding of SecB is not affected and partially allowed an interaction with G3p as shown by size exclusion chromatography (Appendix A).

The translocation of the 379-residue-long periplasmic region of G3p involves the SecA motor component of the translocase. This was expected, since protein regions longer than 80 residues require SecA [37,38]. The SecA protein uses the hydrolysis of ATP to power the movement of the protein chain across the translocase in the membrane [39]. Since this is a linear movement of a protein chain through the translocation pore of SecY, the folding of the G3p N1 and N2 domains has to occur in the periplasm. To ensure that the folding process does not happen already in the cytoplasm, the membrane translocation is most likely initiated co-translationally. Indeed, nascent chains of G3p with the signal sequence exposed out of the ribosome were found to crosslink SecY, the main component of the SecYEG translocase (Figure 5), at positions where signal sequences of other precursor proteins had been found to interact [40].

Our data clearly show that in addition to the SecYEG translocase, the membrane insertase YidC is involved in the translocation of G3p (Figure 4C). Whereas in the initial event as a nascent chain G3p did only show efficient interaction with SecY (Figure 5; Appendix A), further translocation and the insertion of the membrane anchor was clearly accompanied with contacts to the hydrophobic slide of YidC (Figure 6). Several contacts to TM3 (residue 427) and TM5 (residues 501, 505) were documented. The amino acid residue 427 of YidC is intriguingly the same residue that contacts the major coat protein, M13 procoat, during membrane insertion [41]. Particularly, the proteins that use the YidC-only pathway for membrane insertion use the hydrophobic slide consisting of TM3 and TM5. These are the ATP synthase subunit c [42], the mechanosensitive channel protein MscL [43] and the SciP protein of the T6SS secretion system of *E. coli* [44]. We were surprised that we did not find crosslink products of the membrane anchor region with SecY, except for the residue 408 in TM10 (Figure 6A). It was previously shown that this residue is in close contact with the mature part of the translocating proOmpA protein chain (Cannon et al., 2005). Therefore, one explanation is that the translocating G3p protein chain is transfered from SecYEG to YidC when the membrane anchor region is inserted, which occurs predominantly by YidC. SecY TM10 may contact the hydrophobic slide of YidC to hand over the substrate chain. However, YidC is also involved in the transfer of the periplasmic part of G3p, since our protease-mapping results clearly show that in the absence of YidC the periplasmic part accumulates in the cytoplasm and is protected from proteolysis (Figure 4C). Nevertheless, we do not know why SecYEG on its own is not able to transfer the protein chain of G3p into the periplasm if YidC is absent.

## 5. Conclusions

The five coat proteins of bacteriophage M13 are all inserted into the inner membrane of the host before they assemble into a phage particle. The assembly process starts in the membrane with G7p and G9p, two small single-spanning membrane proteins of 33 and 32 residues, respectively. They are synthesised without a signal peptide and possibly insert without an insertase [45], and form a cap structure at the tip of the filamentous phage particle. The assembly process is terminated by G3p and G6p. Although G6p consists only of 112 amino acid residues, it has been predicted to span the membrane three times [46]. Together with G3p, it binds to the end of the filament as G3p copurified with G6p from phage, verifying their molecular interaction [47]. The major coat protein G8p is present on each phage particle with about 2750 copies, whereas the other, minor coat proteins are in about five copies each. Considering the number and speed of the secreted phage from a single host cell [48], the major coat protein must have an enormous production and insertion rate. This might also explain why the G8p procoat protein uses the “YidC-only” pathway for insertion, since this system is very efficient and fast [49] and outnumbers SecYEG [50]. In contrast, G3p uses YidC and SecYEG. We speculate that the low copy number of G3p per phage particle allows a slower membrane insertion of the minor coat protein.

## Figures and Tables

**Figure 2 viruses-13-01414-f002:**
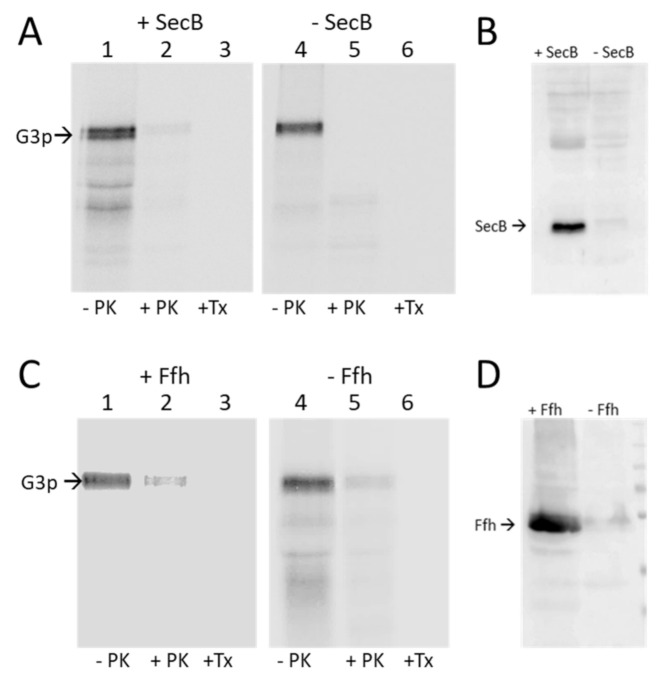
Targeting of G3p is independent of SecB and SRP. (**A**,**B**) The plasmid encoding G3p was transformed into the SecB deletion strain JW3584. (**A**) Exponentially growing cells were induced for 10 min with 1 mM IPTG, pulse labelled with ^35^S-methionine/cysteine for 3 min and converted to spheroplasts (-PK). A portion was digested with proteinase K (+PK) or with proteinase K in the presence of Triton-X-100 (+TX). The samples were immunoprecipitated with an M13 antibody and analysed by PAGE and phosphorimaging. (**B**) Western blot of the JW3584 culture and the control strain MC1061 detecting SecB. (**C**) The araBAD-controlled MC-dFfh strain was used to determine the requirement of SRP for the membrane insertion of G3p. The cells were grown either in the presence of arabinose (+Ffh) or in the absence of arabinose (-Ffh) and treated as described in (**A**). (**D**) Western blot to evaluate the cellular Ffh content in the culture grown with or without arabinose, respectively.

**Figure 3 viruses-13-01414-f003:**
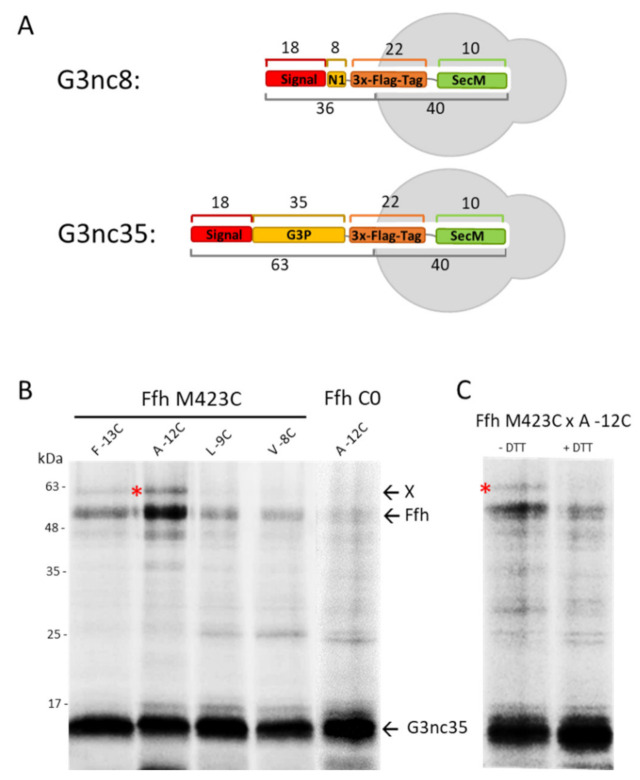
Interaction of G3p nascent chain with SRP in vivo. (**A**) A ribosome-nascent chain (RNC) was constructed with G3p encompassing the signal sequence and the first 35 amino acid residues of the mature protein, followed by a sequence encoding a triple Flag-tag (22 residues) and a SecM stalling sequence of 10 residues. (**B**) Coexpression of the nascent chain G3nc35 and Ffh in *E. coli* MC-dFfh. The exponentially growing cells were pulse-labelled with ^35^S-methionine/cysteine for 2 min and immunoprecipitated with a Flag-tag antibody. Single cysteine mutants at the signal sequence positions −13, −12, −9, −8 of G3nc35 were coexpressed with Ffh423C or Ffh without cysteine for a control. (**C**) Coexpression of G3nc35(-12C) with Ffh423C as in (**A**), analysed on PAGE in the absence or presence of dithiothreitol (DTT). The crosslinked G3p-Ffh product is marked with red asterisks.

**Figure 4 viruses-13-01414-f004:**
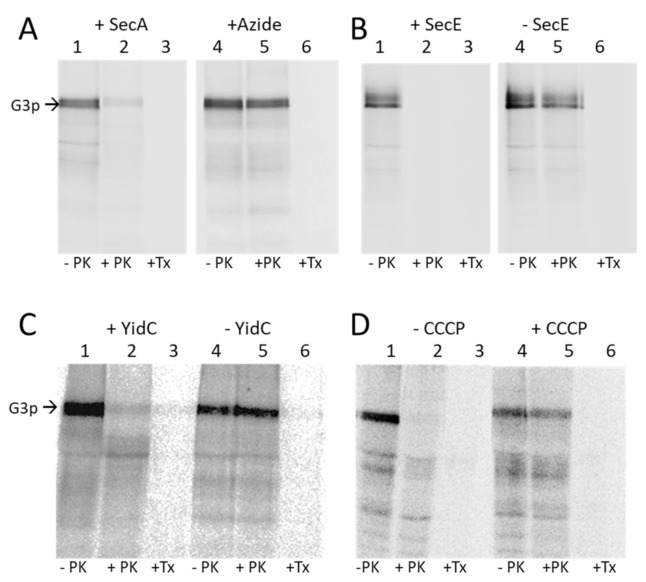
Membrane insertion of G3p requires SecAYEG and YidC and the electrochemical membrane potential. (**A**) The insertion of G3p was tested in MC1061 (lanes 1–3) and in azide-treated MC1061 (lanes 4–6). The exponentially growing cells expressing G3p were pulse labelled with ^35^S-methionine/cysteine for 3 min and chased with L-methionine/cysteine for 2 min, converted to spheroplasts (lane 1, 4) and digested with proteinase K (lanes 2, 5) or treated with Triton-X-100 and then protease digested (lanes 3, 6). The cells were acid-precipitated, immunoprecipitated and analysed by PAGE and phosphorimaging. (**B**) MG1655 cells were grown in the presence of arabinose, which allows expression of SecE (lanes 1–3), or in the absence of arabinose, which depletes the SecE (lanes 4–6), and treated as described for (**A**). (**C**) MK6 cells grown in the presence of arabinose expressing YidC (lanes 1–3) or in the absence of arabinose that depletes YidC (lanes 4–6) were treated as described for (**A**). (**D**) MC1061 cells (lanes 1–3) and CCCP-treated MC1061cells (lanes 4–6) expressing G3p were analysed for G3p insertion by protease mapping as described for (**A**).

**Figure 5 viruses-13-01414-f005:**
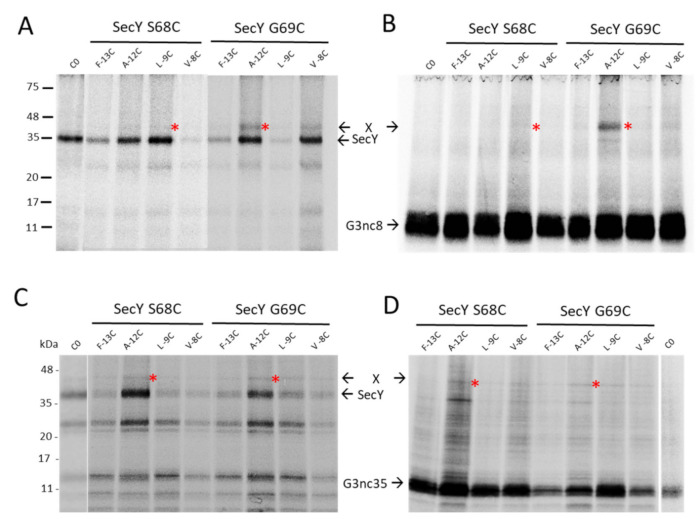
Ribosome-nascent chains (RNCs) of G3p contact SecY. The plasmids encoding G3nc8 (**A**,**B**) and G3nc35 (**C**,**D**) were coexpressed with plasmid-encoded SecYEG in *E. coli* MG1655-SecE. The G3ncs had single cysteine mutations at positions −13, −12, −9 or −8 in the signal sequence, and in SecY a single cysteine at position 68 or 69, respectively. The exponentially growing cells were induced with 1 mM IPTG for 10 min, pulse labelled with ^35^S-methionine/cysteine for 2 min, treated with copper phenanthroline for 10 min and processed for immunoprecipitation with an antibody to the T7-tag recognizing SecY (**A**,**C**) or to the Flag-tag recognizing the G3nc (**B**,**D**). For a control, a plasmid expressing the cysteine-less SecY was analysed. The potential crosslink products are marked with red asterisks.

**Figure 6 viruses-13-01414-f006:**
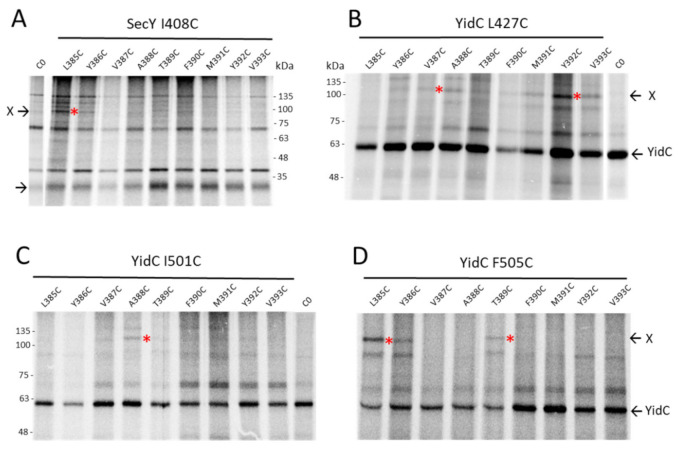
The transmembrane anchor region of G3p interacts with SecY and YidC. (**A**–**D**) The core of the G3p membrane anchor (residues 385–393) was scanned for interactions by single cysteines at these positions. (**A**) Coexpression of the G3p mutants with SecYEG with a single cysteine at position 408 in TM10 of SecY in *E. coli* MG1655-SecE. The cells were treated as described in Figure 5 but immunoprecipitated with an antibody to the T7-tag. (**B**–**D**) Coexpression of the G3p mutants with YidC in *E. coli* MK6 grown without arabinose. The plasmid-encoded YidC had a cysteine at residue 427 in TM3 (B), 501 (**C**) or 505 (**D**) in TM5, respectively. The cells were treated as described in Figure 5 but immunoprecipitated with an YidC antiserum. The potential crosslink products are marked with red asterisks.

**Figure 7 viruses-13-01414-f007:**
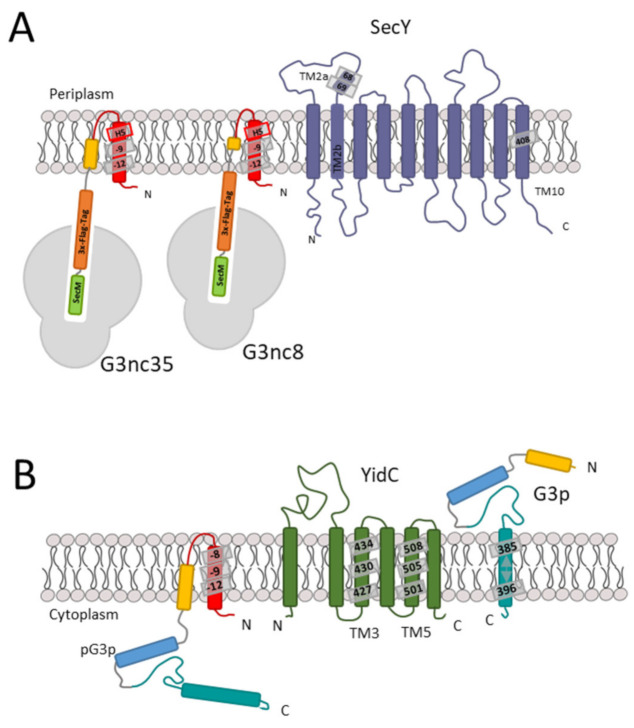
Stages of membrane insertion of G3p. (**A**) In the first stage, when the nascent chain of G3p exposing the signal sequence interacts with SecY of the translocase, namely in the plug region, the residues 68 and 69 are predominant. (**B**) In the second stage, the periplasmic region of G3p is translocated by SecA and SecYEG. The membrane anchor region of G3p is then inserted by contacting the YidC hydrophobic slide residues 427, 501 and 505.

## Data Availability

We want to note that we followed the MDPI Research Data Policies at https://www.mdpi.com/ethics.

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
