# Peer review of "Membrane Insertion of the M13 Minor Coat Protein G3p Is Dependent on YidC and the SecAYEG Translocase"

_viruses, 2021, doi:10.3390/v13071414_

Round 1
Reviewer 1 Report
Drs Kleinbeck and Kuhn studied the biogenesis of the minor M13 coat protein G3p in E. coli cells, specifically the components and mechanism of its membrane insertion. The two authors employed E. coli strains that lack the components of interest one by one as well as disulphide crosslinking between components and nascent G3p chains.
The paper shows that G3p depends on SecA plus SecYEG complex as well as YidC for its membrane insertion. Furthermore, the cross linking demonstrates that G3p can interact with SRP in the cytoplasm prior to its membrane insertion and that the order of events during the actual membrane insertion process is that SecA plus SecYEG are required for membrane transfer of the periplasmic domain of G3p and that YidC is involved in integration of the membrane anchor into the bacterial inner membrane.
The paper is written well and nicely describes the M13 G3p biogenesis in E. coli prior to phage assembly. In principle, it can be published as it is.
I have only a couple of minor suggestions for improvement:
1. In my view, the wording of both Abstract (lines 8-10) and Introduction (lines 55-57) give the impression that it had previously been shown that G3p biogenesis involves SecYEG plus YidC. To my knowledge, however, that is not the case and is shown here for the first time.
2. I wonder if the authors also addressed the suggested interaction of G3p with SecB by crosslinking. A statement towards this point should be added to section 3.2 or the Discussion.
3. There appears to be too much space after the full stop in line 254; in line 273 a space is missing between 1 and h.
Author Response
Answers to reviewer 1
- We changed the respective sentences in the abstract.
- Actually, we tested the binding of G3p to SecB by size exclusion chromatography indicating an interaction. The data are now included in the new Fig.S5.
- We repaired the respective errors.
Reviewer 2 Report
The molecular details of the M13 phage infection process in E.coli and the subsequent assembly of phage progeny particles remain poorly understood. The present article concerns with the question how the newly synthesized minor M13 phage coat protein G3p is targeting and inserting into the inner E,coli membrane. Using thiol crosslinking experiments, the authors showed that during protein biosynthesis the ribosome-exposed leader sequence of the nascent G3p first contacts SecY. During progress of protein biosynthesis, the hydrophobic C-terminal membrane anchor region interacts with YidC. This is an important contribution resolving the role of the membrane insertion process of the M13 G3p coat proteins for the final phage assembly process.
The manuscript is very well presented and should be published without further modification.
Ref 1: Would be good to cite an additional reference on bacteriophages which is easily accessible for a nonspecialised reader.
Fure 1, legend, last sentence: There is a typo. It should read "Depicted are in the outer membrane OmpA ..."
Author Response
Answers to reviewer 2
- Reference is changed to Loh et al.
- We repaired this error.